# Three-State Opinion Model on Complex Topologies

**DOI:** 10.3390/e24111627

**Published:** 2022-11-10

**Authors:** Irene Ferri, Conrad Pérez-Vicente, Matteo Palassini, Albert Díaz-Guilera

**Affiliations:** 1Departament de Física de la Matéria Condensada, Universitat de Barcelona, 08028 Barcelona, Spain; 2Universitat de Barcelona Institute of Complex Systems (UBICS), Universitat de Barcelona, 08028 Barcelona, Spain

**Keywords:** sociophysics, opinion dynamics, magnetic models, complex networks, community structure

## Abstract

We investigate opinion diffusion on complex networks and the interplay between the existence of neutral opinion states and non-trivial network structures. For this purpose, we apply a three-state opinion model based on magnetic-like interactions to modular complex networks, both synthetic and real networks extracted from Twitter. The model allows for tuning the contribution of neutral agents using a neutrality parameter. We also consider social agitation, encoded as a temperature, that accounts for random opinion changes that are beyond the agent neighborhood opinion state. Using this model, we study which topological features influence the formation of consensus, bipartidism, or fragmentation of opinions in three parties, and how the neutrality parameter and the temperature interplay with the network structure.

## 1. Introduction

For the past few years, sociophysicists have been developing new tools in order to understand the main mechanisms underlying large-scale human behavior. The main goal in the field is to achieve a deeper understanding of how society functions and some degree of predictive power. These tools can be used to detect flaws in social organizations or decision-making regarding specific questions [1].

Human societies are complex and diverse, and exhibit a large variety of behaviors. Rather than developing a single model that aims to explain human behavior in a global manner, we can approach different types of questions by developing specific models that isolate the main characteristics of the question of study. Establishing connections between them by analyzing their main findings may lead to develop a more general framework for human behavior description [2,3,4].

Regarding opinion dynamics, a collection of models have been proposed and applied so far. Some of them introduce the opinion space as a discrete variable which can take two or more values [5,6], while others instead propose that opinion can take any value within a continuous interval [7,8,9,10].

Despite the differences in the details of each model, the questions they intend to answer are mainly related to the distributions of opinions in the population. In the attempt to model basic human behavior, the complex systems community realized very soon that the analogies with well-known physical models could be exploited. For instance, the tendency to align one’s opinion with those of their neighbors is a property that ferromagnetic materials exhibit as well [11,12,13]. In fact, the Ising model has been extensively studied in the context of binary opinion analysis [14,15,16,17], which, in turn, motivates the study of this model on new topologies, such as synthetic graphs and real networks [18,19]. Embedding the model on complex topologies has expanded the knowledge about fundamental questions regarding critical behavior and transitions between states. At the same time, the study of the dynamics of these models on networks can also help in solving complex systems problems, as for instance community detection [20] or brain functioning [21,22,23].

The voter model, which in some cases has a mapping to the Ising model, is another paradigmatic reference in opinion dynamics, again for binary opinion spaces [5,24]. The Potts model has also been applied to social physics in order to offer a broader opinion space which takes into account three or more opinions and can be more suitable for modeling non-polarized situations, in which a third state does not imply an intermediate opinion between two others [25,26].

Another possible scenario is the three-party scene, which includes extremists with two opposite opinions and a neutral fraction of individuals whose opinion can be placed in between these two sides. Neutral agents have been found to be relevant for many studies [27,28,29], since a lot of social debates can be framed as a three-party dialogue. For instance, during the Catalan popular consultation in 2014, there were three possible answers to the independence question: “Catalonia should become an independent state”, “Catalonia should not become an independent state”, and an intermediate response “Catalonia should become a state, but not be independent”, which could be associated with a neutral position. Another example could be language competition in a multilingual social context, which has also been discussed in the literature as a three-state problem [30]. For instance, the authors of Ref. [27] propose a three-party Ising-like model where extremists are incompatible and do not affect each other, thus the only opinion changes allowed are those involving neutral agents, which often leads to situations with a mixture of extremists. Our model, also inspired by magnetic interactions, considers the same three opinion states for the agents and allows transitions between all of them. We introduce a third opinion, which, in terms of distance, is placed symmetrically between the two usual (polarized) ones, and we quantify the probability transition to this state by means of a neutrality parameter.

Working within this framework, we aim to explore the impact of the neutral opinion on the absorbing states of the system. Additionally, we consider the existence of a social temperature, which is intended to model non-homophilic opinion changes. We present elsewhere [31] a detailed analysis of the model on the fully-connected and Erdös-Rényi graphs. In the present work, we apply the model to other topologies, including real networks extracted from Twitter.

The paper is organized as follows: In Section 2, we describe the model and methods, and, in Section 3, we summarize the main results of the analysis performed in [31], apply the model to modular synthetic networks, and present results on real ones extracted from Twitter data. Finally, in Section 4, we expose the main conclusions of the study.

## 2. Materials and Methods

We propose a model with *N* agents embedded on a generic, undirected and unweighted graph. Each agent can be in one of the following individual opinion states, represented by two-dimensional vectors:$S_{i}=\left(1,0\right)$; positive opinion/rightist;$S_{i}=\left(0,\alpha \right)$; neutral opinion/centrist;$S_{i}=\left(-1,0\right)$; negative opinion/leftist,where α is a dimensionless parameter and the index i=1,⋯,N refers to any agent in the system. Interaction is considered to be in pairs and tends to minimize the following Ising-like Hamiltonian: (1)$$H=-J\sum_{\langle i,j\rangle }S_{i}\cdot S_{j},$$
being the sum extended over nearest neighbors. Therefore, every pair of agents connected by an edge contribute to the internal energy of the system. We calculate each contribution as the scalar product of the two opinion vectors, and we sum all the contributions to compute the global internal energy (see Figure 1). The coupling constant *J* is positive, and we take it equal to one for simplicity. As the Hamiltonian is defined over nearest neighbors, its form reflects the fact that having an opinion different from those of the people you are connected with has a cost, while agreement with neighbors decreases the system energy asymmetrically for extremists and neutral agents. By regulating the parameter α, we study the effects of the neutral population on the state of the system.

The ground state of the system corresponds to consensus in one polarized opinion for α<1, to neutral consensus for α>1 and to consensus in any of the three opinions for α=1 [31]. We consider that the system is in contact with a thermal bath, here temperature being understood as a mechanism for opinion adoption different from those captured by Equation (Equation 1). While the Hamiltonian rewards that linked agents share the same opinion, the thermal bath gives the agents a certain probability for adopting an opinion that increases the system energy. This behavior can be interpreted as social agitation or free-will [29,32,33].

In order to compare graphs with different average connectivity 〈k〉≡z in a meaningful way, we always consider the rescaled reduced temperatures $T^{*}=T/z$. The neutrality parameter α regulates the contribution of neutral neighbors to the energy of the system. The order parameters are the difference between the fraction of rightists and leftists, $m=(N_{+}-N-)/N$, called magnetization due to its analogy with magnetic systems, and the fraction of neutral agents $n_{0}=N_{0}/N$.

We use simulations in order to explore all topologies described in the following section. We set initial opinions uniformly at random, as we suppose that a priori each agent could be in any opinion state. At each step, the Metropolis Monte Carlo (MMC) algorithm we use selects one agent at random, and it proposes a random opinion change. The proposal will be accepted with a probability one if this change does not increase the system energy. Even if the opinion change we propose increases the energy by a certain amount ΔH, it still can be accepted with a probability $e^{-\Delta H/k_{B}T}$. The greater the temperature, the easier it is for the agent to disregard their neighbors’ opinion during the interaction. Simulations run until the system energy stabilizes.

## 3. Results and Discussion

### 3.1. Previous Results

It was shown in [31] that the equilibrium mean-field phase diagram of the model defined above presents both a discontinuous and a continuous transition, separated by a tricritical point $\alpha_{tc}$. When α is below $\alpha_{tc}$, the magnetization tends to one (polarized consensus) when T→0 and decreases continuously with the temperature until it reaches the value 0 at $T=T_{c}\left(\alpha \right)$. For $\alpha_{tc}<\alpha <1$, the magnetization jumps discontinuously to zero at a temperature $T=T_{d}\left(\alpha \right)$. The two lines $T_{c}\left(\alpha \right)$, $T_{d}\left(\alpha \right)$ merge at the tricritical point. The fraction of neutral agents is zero when T→0 and has a peak located at $T_{c}\left(\alpha \right)$ that becomes a jump at the discontinuous transition for $\alpha_{tc}<\alpha <1$, then it decays towards 1/3 when T→∞.

The zero-temperature dynamics on complete graphs has also been studied in [31]. It was found that, despite the ground state being the polarized consensus for α<1, the neutral consensus becomes an attractor in the range $\left[\alpha_{tc},\infty \right)$. This is true for a large range of initial conditions, including cases with a random uniform distribution. However, when the system is slightly heated, it overcomes the energetic barriers between the local and the global minimum and then it falls into the polarized consensus.

Many features of the mean-field solution are found in random graphs as well. However, unlike in the fully connected graph, where the absorbing state at low temperatures is always consensus, in random graphs with low average connectivity z≲4, agents can get trapped into local energy minima, which correspond to an ensemble of non-consensus isoenergetic configurations with small deviations in the number of agents in each state. This behavior, which is observed for all values of $\alpha \in \left[0,1.25\right]$, is caused by the presence of blinkers, nodes that lie in the middle of two sub-communities with opposite opinions and keep changing their state indefinitely with no energy cost. The presence of blinkers practically vanishes for z>4; therefore, random networks with a connectivity above this value reach consensus at low temperatures [31].

### 3.2. Modular Networks

Our purpose now is to assess the role of community structure in the opinion dynamics of our model. In the limit of low temperatures, the homophilic term that drives the agents to agree with their neighbors is expected to dominate over the social agitation. Therefore, we put the focus on the achievement of a polarized or a neutral consensus, or on the contrary the arising of bipartidism or tripartidism at low levels of temperature. We explore the outcomes for different levels of the neutrality parameter α.

Many complex networks, and social networks in particular, have been shown to have a clear community (modular) structure [34]. Community detection has been intensely studied and many different methods have been suggested [35]. At the same time, benchmarks for testing the efficiency of these methods have also appeared in the literature. Among those, one of the the most commonly used was the one proposed by Newman and Girvan [36]. There, the authors construct a set of networks with different community structures. Each network has 128 nodes divided into four communities of 32 nodes each. In the original model, links are established independently at random between nodes with probability $p_{in}$ if both nodes belong to the same community and $p_{out}$ otherwise, with z=16. Here, we work with a slightly different version in which we fix the number of links a node has to nodes in the same community, $k_{in}$, and to other communities, $k_{out}=16-k_{in}$. In this way, we can tune the relevance of the community structure, which is evaluated in terms of the modularity.

In Figure 2a, we can see the average of the absolute value of the magnetization as a function of the number of intra-community links at a very low temperature for different values of α. The green line corresponds to the modularity of the best partition. There is a clear change in behavior around $k_{in}=8$ that can be understood by examining the probability of acceptance for a given flip proposal in the dynamics. In particular, if we consider two communities of agents, each holding +1 and −1 opinion, respectively, we have that, for α=0, the probability of an agent in the positive community to change opinion to −1 (in two steps, i.e., passing through the neutral opinion) is equal to $P_{acc}=\min\left[\exp(\left(16-2\cdot k_{in}\right)/T^{*}),1\right]$, where $k_{in}$ denotes the number of intra-community links. Therefore, when $k_{in}\leq 8$, we have $P_{acc}=1\;\forall T$.

For low values of α, these networks are not able to achieve global consensus when the number of intra-community links exceeds 8 (however, the modules reach internal consensus in all cases; see Figure 3a). When α=0.75, we observe that 〈|m|〉≠1 even for $k_{in}\leq 8$. We speculate that these results are related to a putative first order transition analogous to that of the fully-connected graph [31], which may occur at alpha around the interval α∈(0.8,1). In this range, the dynamics end up in consensus, but this consensus occurs in a polarized opinion for roughly half of the simulations, and in a neutral opinion the other half. For $k_{in}>8$, we observe a decay in 〈|m|〉, similar to the one presented for α=0. Finally, for α≥1, when the neutral consensus becomes an absolute energy minimum, we observe clear preference for the neutral consensus for all $k_{in}$. Nevertheless, some polarized clusters can appear for α=1, especially for the largest values of intra-community links.

Figure 2b shows the average value of the magnetization as a function of temperature for several values of intracommunity links $k_{in}$ and α=0. For values $k_{in}>8$, we observe a peak which indicates that the greatest majority in a polarized opinion occurs at T>0. As we mentioned above, the different communities arrive at a local consensus in the steady state, but, in general, the modules do not share all the same opinion. Figure 2c shows the position of this peak for every value of $k_{in}$. Above $k_{in}=8$, the system still reaches |m|∼1 at a temperature that increases with the number of intra-community links until $k_{in}=11$; above this value, the maximum value of 〈|m|〉 starts decreasing.

The case $k_{in}=15$, with very well-defined communities in the limit of low temperatures, is examined in Figure 3. The left panel shows an example of a final configuration for this network at $T^{*}=0.1$ and α=0.75. This is just one of the possible final outcomes for this network, in which magnetization and fraction of neutral agents turn out to be |m|=0 and $n_{0}=0.5$, respectively. The right panel shows the possible final values for |m| and $n_{0}$ for 30 MMC repetitions, starting from random initial conditions. These results show that the final configurations correspond to situations of consensus within communities but, in general, the system does not reach a global consensus. Notice that neutral communities do not have representation for α≲0.5; therefore, in this range, the final $n_{0}$ is always 0, and the magnetization can take three possible values:|m|=0, corresponding to a system divided into two communities, each one holding a different extremist opinion.|m|=0.5 that is obtained when three communities hold the same polarized opinion and the fourth holds the opposite one.|m|=1, if by chance the system reaches the global consensus.

When 0.5≲α≲1.0, the number of possible final configurations is larger because they include all combinations with communities in any of the three opinion states. For α≳1, the contribution of the neutral nodes to the energy becomes larger than the contribution of polarized agents and most often the system reaches neutral consensus, characterized by |m|=0 and $n_{0}=1$. However, in some cases, the magnetization takes the value |m|=0.25 and the fraction of neutral agents is $n_{0}=0.75$, indicating that one extremist community appears in the final state while the other three are neutral. Finally, for values of α≳1.4, we always obtain neutral consensus as the final macrostate.

We have also considered synthetic modular networks with a hierarchical community structure formed by more than one cluster level. These networks have been used in previous studies regarding community detection [37], in this case using the Kuramoto model, as they are specifically constructed to highlight the nodes correlations between the final states of a specific dynamics.

Models based on magnetic-like interactions, as the Ising or the Potts model, have been used previously for community detection [26,38,39,40]. In our case, it turns out that the correlations of the final opinion state of the nodes are sensitive to the value of α. Actually, for small α, only the smaller clusters arrive to local consensus; however, larger values of α enable consensus in a larger scale. We can take advantage of this feature and reveal the different levels of structure by tuning the parameter α, like other multiresolution methods [41] have done before. In this way, we can study at the same time the opinion dynamics on a particular network and learn about its community structure.

In practice, what we do is to define a matrix ${\hat{C}}_{ij}$ whose elements account for the number of times the nodes *i* and *j* end up holding the same opinion in the simulation. We run a large number of simulations $N_{reps}$ and normalize this correlation value as $C_{ij}={\hat{C}}_{ij}/N_{reps}$. As our Hamiltonian (Equation 1) only contains positive interactions between nodes (i.e., there is no repulsive term which drives neighbors to have different opinions at low temperatures), we expect a value $C_{ij}=0$ for pairs of nodes which belong to different, well segregated communities.

In Figure 4, we can see a network with two community levels generated as follows: a set of 256 nodes is divided into 16 clusters that will represent the first community level. The second organizational level of the network is formed by four compartments, each one containing four different clusters of the first level. Here, node colors do not represent opinions but are just to clarify the network topology. The results for α=0 show clearly the first level, corresponding to the smallest subgraphs that appear with correlation $C_{ij}=1$ in 16 × 16 boxes in the main diagonal (Figure 4b) (We expect that, for α=0, the results would resemble those obtained by using the Ising model). The second level is not so evident, but, when we use α=0.75, we can distinguish it better, as the correlations within the four big compartments are stronger. This property, which is caused by the fact that a higher value of the neutrality parameter increases opinion diffusion, can also be used to detect asymmetries in the link distribution between nodes. For example, in Figure 5, we can see a network formed by four compartments which are in turn divided in two subgroups, but connections are not perfectly symmetric, unlike in the case shown in Figure 4. In particular, we can see one node, marked with a circle in Figure 5a that has more out-community links outside its compartment (on the second level of community structure) than the rest of the vertices. This feature is not visible when we perform simulations with α=0, but it becomes noticeable when we use α=0.75.

### 3.3. Real Networks

We started applying the model on well-known topologies that have been used for benchmarking reasons [35]. Let us now analyze its behavior when embedded on real social networks. In particular, we study interaction networks of Twitter handles around a given topic, i.e., hashtag. Each vertex in the network represents a Twitter handle and a link represents an interaction (retweet or mention) between two handles in a tweet containing the selected hashtag. Multiple edges and self-edges have been removed from the network, as well as small non-connected components.

Data (provided by Associació Heurística) were collected through the Twitter Standard Search API, which returns a collection of Tweets matching a specified query, namely a hashtag or a set of keywords. The network was built by adding an edge between two Twitter users whenever a user retweets or mentions another one. The fact that they are built from retweets/mentions instead of “follows” is the reason for the low clustering coefficient of these networks, dominated by star-like structures.

Twitter networks are directed, since one can follow a user that does not follow you back; however, we have considered links as undirected, for simplicity. The graph is not used for a realistic embedding but rather as a proxy for social networks of information diffusion. Another hard assumption is the achievement of a stationary state; surely, many observable states are transient, since social systems are often perturbed by supervening events, but this is beyond the scope of the present work.

The networks we chose correspond to the hashtags #yotambiensoynazi (which in English means I am also a Nazi, and we shorten as #yotambien), #nochebuena (in English Christmas Eve) and #martarovira (the name of a Catalan politician). (The first one is a small network that does not have to be confused with the feminist movement since it actually corresponds to an altercation occurred in Zaragoza (Spain), in the context of the Catalan independence process. The second one came in the wake of a speech that the Spanish King delivered against the aforementioned process on 3 October 2017. This official communication was sarcastically criticized by pro-independence supporters, making fun of it by using a comparison with the King’s yearly Christmas Eve speech. The third one is associated with the Spain government legal actions against the politician Marta Rovira, again in the context of the independence process in Catalonia.)

Some properties of these networks are displayed in Table 1. The table shows that the absolute value of the magnetization at very low *T* is inversely proportional to the modularity value of the best partition.

Despite not considering transient regimes and directed or weighted links, we would expect to be able to observe some well known social phenomena regarding opinion spreading. According to simulations, all of these systems are unable to reach consensus at low temperatures, as we see in Figure 6. The behavior resembles the one found in BA networks with 〈k〉=2, indicating that the dynamics are mainly driven by the existence of highly connected vertices which lead to the formation of opinion bubbles (Figure 7) because the opinion of the hubs is harder to change. The attractors of the dynamics at low temperature are metastable configurations that depend on the particular initial configuration, which is set at random for every repetition; therefore, we observe a large dispersion in the results. Blinkers, which are present for instance in random networks with low connectivity (see Section 2), appear in the final configurations reached by the MMC simulations for these networks as well.

In order to infer which topological property is related to the lack of consensus at T→0, we have compared different network coefficients and the value of 〈|m|〉 at T=0 for α=0.5 (see Table 1) for the three real networks, the BA graph with 〈k〉=2 and the ER network with 〈k〉=4 (included as a null model). The comparison suggests that that the maximum value of the magnetization that the system is capable of reaching is related to the highest modularity value of each network.

Note that the strong modular structure prevents the system from reaching zero-magnetization at T→0, even for α=1.25, where the energy has a unique global minimum at the macrostate of neutral consensus. As in some networks analyzed in the previous section, once opinion bubbles are formed, changes between different microstates become too costly, not only between extremists but also when the change occurs from or towards the neutral opinion.

The highest value for the magnetization at low temperatures is found at α around 0.5; α=0.75 in the case of BA graphs with 〈k〉=2. When the magnetization reaches its maximum value the fraction of neutral agents $n_{0}$ is zero. From a social point of view, this means that the system gets divided into communities formed by positive or negative agents, but there is a stronger majority in one polarized opinion than for other values of α. Neutral communities are found at higher values of α, destabilizing the polarized majority. The fact that a moderate neutral intensity catalyzes the majority in one polarized opinion adds to the conclusions presented in [42].

The networks #martarovira and #nochebuena exhibit a non-monotonic behavior of 〈|m|〉 with *T* for α={0,0.5}. This is caused by the fact that the model presents a very complex energy landscape with multiple local minima when embedded in these topologies. This landscape is sensitive to the neutrality parameter α, since it changes the energy difference between the local neutral consensus and local polarized consensus. When the temperature increases, some energetic barriers can be easily overcome, so the attractors of the dynamics change. In some cases, this can enhance the dominance of one polarized opinion over the other one, which corresponds to higher average absolute magnetization; in others, it is vice versa.

The community structure of real networks can be easily visualized using the method described in the Appendix A. In Figure 8a, we observe a large community containing approximately half of the nodes and a few smaller clusters. Simulations using α=0 and α=0.75 show just some minor differences in this network; for instance, the biggest community is slightly smaller. The network does not appear to have more than one level of community structure.

## 4. Conclusions

Capturing the essence of opinion dynamics has become one of the topics of computational social science, and three-state opinion models could eventually be applied to a large number of current debates. Here, we apply a three-state model that incorporates a notion of distance between states, to modular networks, both synthetic and real. The model also considers the presence of temperature/social agitation.

At low temperatures, homophilic behavior dominates, and the system reaches either global or local consensus. At very high temperatures, the entropic term becomes dominant and agents change constantly their opinion regardless of their neighbors’ states. This disordered state could be associated with conflict or riots. Although it may seem counterintuitive, moderate levels of social agitation can also help to catalyze consensus in modular graphs. On some networks, the system is unable to reach consensus at T→0, but it can do so when there is enough energy available in the form of upheaval. This energy is used by agents to transition between states until the system reaches a global ordered state (consensus), which is more difficult to destabilize, since it corresponds to an energy minimum.

A modularity value for the best partition M≳0.6 indicates that the network has a well defined community structure, with multiple attractors that exhibit partial order. When this happens, the steady state at low temperatures is not consensus but a population fragmented into opinion clusters, which are different depending on the initial conditions. Some “blinker” agents may appear at the domain walls in some topologies; these nodes freely change their opinion, since they are connected to two or more clusters aligned in different directions. The opinion fragmentation into clusters, known as echo-chambers or opinion bubbles’ formation, is mainly related to the network topology, but the contribution of neutral agents to the energy of the system is also relevant. Within a certain range of the neutrality parameter value that depends on the particular network and is always below α=1, the neutral state becomes a bridge between polarized states. This behavior enhances polarized consensus; since agents do not remain in the neutral state, they just use it to transition between opposite opinions.

However, at higher levels of social agitation, agents can overcome some energetic barriers between domains, which, for α<1, increases the magnetization and hence the system gets closer to consensus. In this situation, if there were some neutral clusters at low temperatures, these tend to disappear when *T* rises. Taking the Newman–Girvan network with $k_{in}=15$ as a benchmark for these cases, neutral clusters appear for α≳0.5. On the other hand, when the best partition of the network has a modularity value M≲0.6, there is not a strong community structure in the system. In these cases, the population of agents already reaches the most possible ordered state at T→0, and its dominant order parameter (either |m| for low α or $n_{0}$ for high α) decreases at intermediate temperatures by the effect of thermal fluctuations.

Opinion formation appears to be strongly determined by the action of influential nodes and modularity. Social media networks exhibit tree-like diffusion properties, which naturally lead, in our model, to bipartidist or tripartidist configurations, depending on the strength of the neutral opinion α. If the modularity of the best partition is low enough, a moderate level of noise allows the agents to partially align. Results in real networks suggest that intermediate levels of individual thinking (the existence of a finite temperature that does not exceed the critical point) can lead closer to consensus than pure imitation of linked agents ideology (T→0).

## Figures and Tables

**Figure 1 entropy-24-01627-f001:**
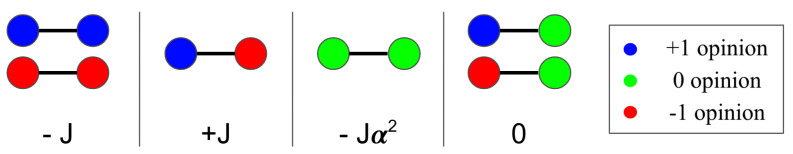
Contributions to the total energy for every possible pair of agents. Colors represent the individual opinion states. J is the coupling factor and α the neutrality parameter.

**Figure 2 entropy-24-01627-f002:**
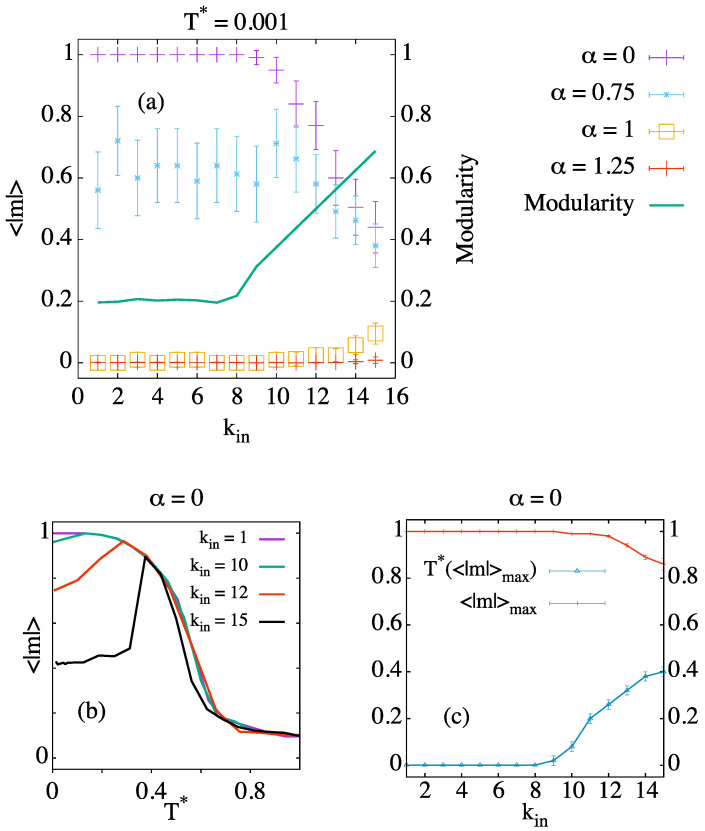
(**a**) Absolute value of the magnetization in the stationary state as a function of the number of intracommunity links at $T^{*}=0.001$, for different values of α; (**b**) absolute value of the magnetization versus temperature for several values of the number of intracommunity links and α=0 (for clarity, errorbars have been removed); (**c**) position of the magnetization peak as a function of the number of intracommunity links. Results are averaged over 100 simulations, except those for $k_{in}=15$ in panel (**b**), which are averaged over 500 simulations because of large fluctuations.

**Figure 3 entropy-24-01627-f003:**
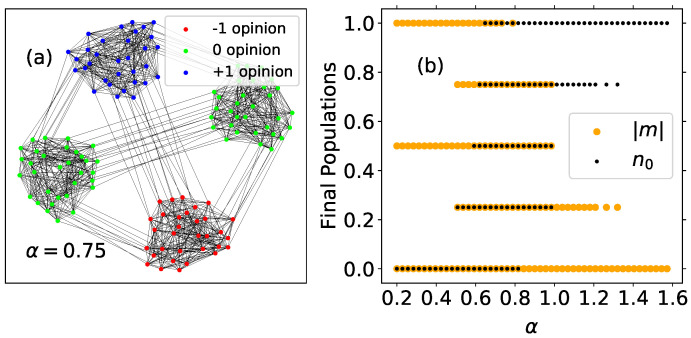
(**a**) Example of a stationary configuration for the $k_{in}=15$ network at $T^{*}=0.1$ and α=0.75, corresponding to m=0 and $n_{0}=0.5$; (**b**) final values of |m| (orange dots) and $n_{0}$ (black dots) for the $k_{in}=15$ network at $T^{*}=0.1$ versus α. Outcomes of 30 independent simulations with $10^{4}$ MMC steps for several values of α between 0.2 and 1.6. A dot is plotted when at least one of the simulations ends up at these values of |m| and $n_{0}$.

**Figure 4 entropy-24-01627-f004:**
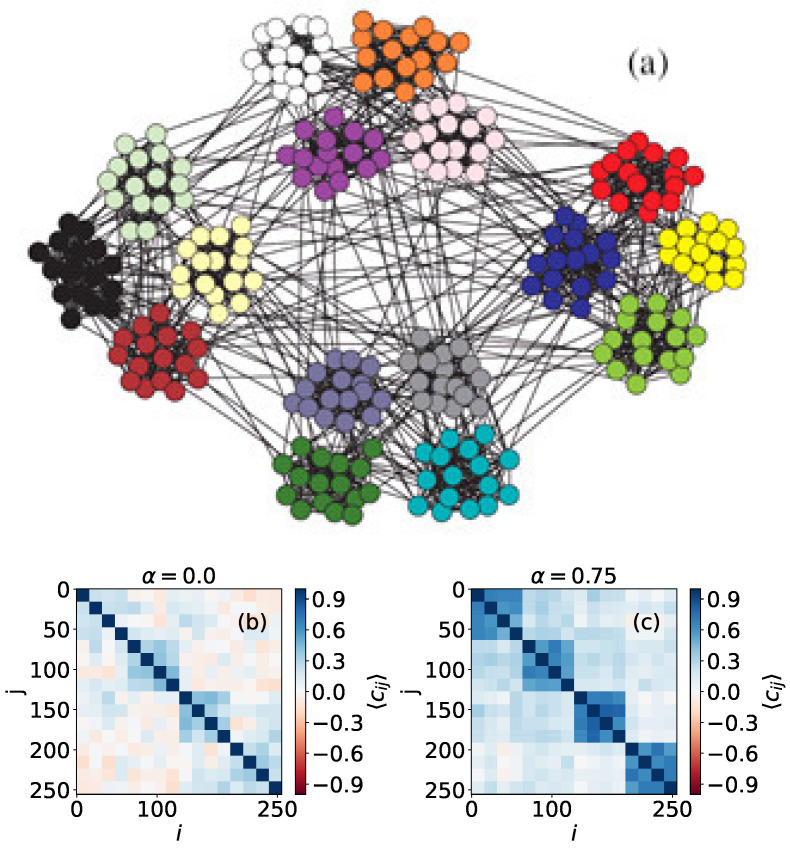
(**a**) Representation of a two-level synthetic network formed by N=256 nodes. Colors are just here to visualize the community levels; (**b**) correlation values for every pair of nodes, obtained using α=0; (**c**) correlation values for every pair of nodes, obtained using α=0.75. Results are obtained using 5000 MMC steps, and are averaged over 200 repetitions.

**Figure 5 entropy-24-01627-f005:**
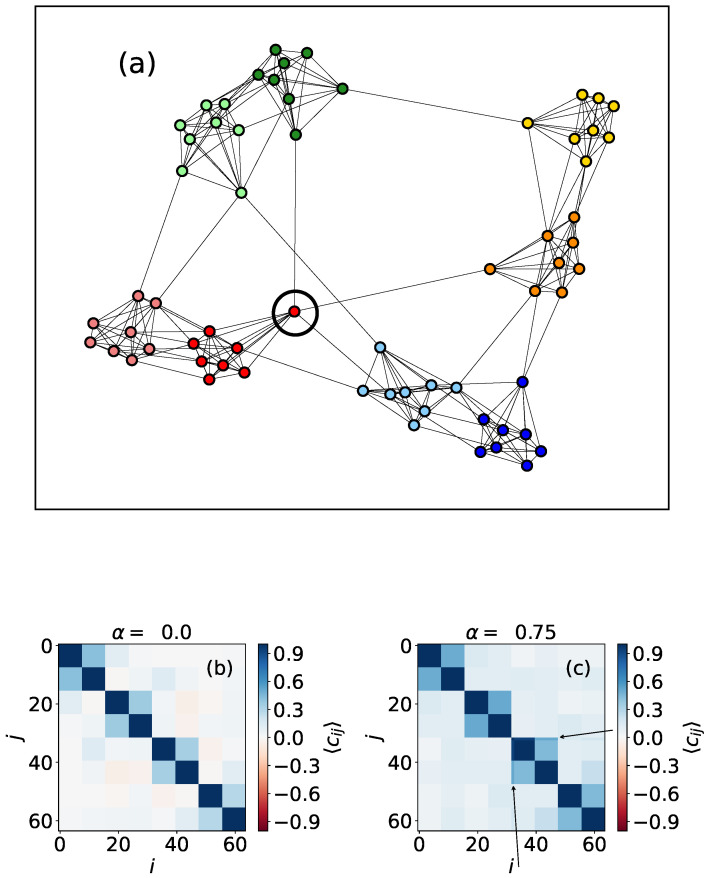
(**a**) Representation of an heterogeneous two-level synthetic network formed by N=64 nodes. Colors are just here to visualize the community levels; (**b**) correlation values for every pair of nodes, obtained using α=0; (**c**) correlation values for every pair of nodes, obtained using α=0.75. Results are obtained using 5000 MMC steps, and are averaged over 1000 repetitions.

**Figure 6 entropy-24-01627-f006:**
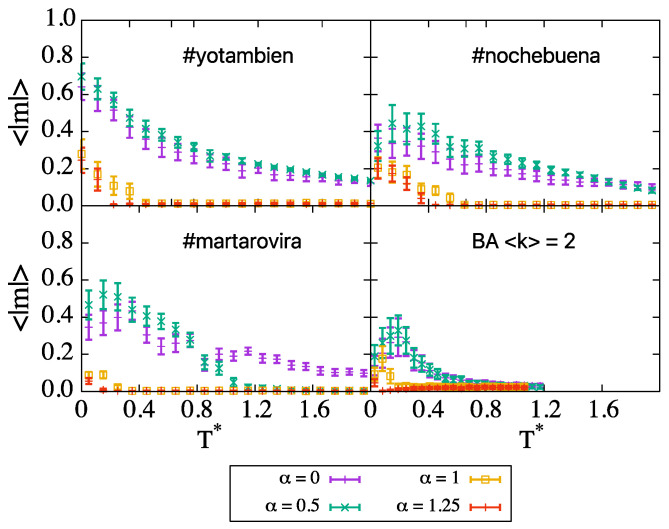
Comparison of the magnetization curves for three real networks and the BA network with 〈k〉=2, all of them presenting frozen configurations at low temperatures, dominated by the presence of big hubs in the network. Results are averaged over 100 repetitions, except for #martarovira, whose results are averaged over 20 repetitions.

**Figure 7 entropy-24-01627-f007:**
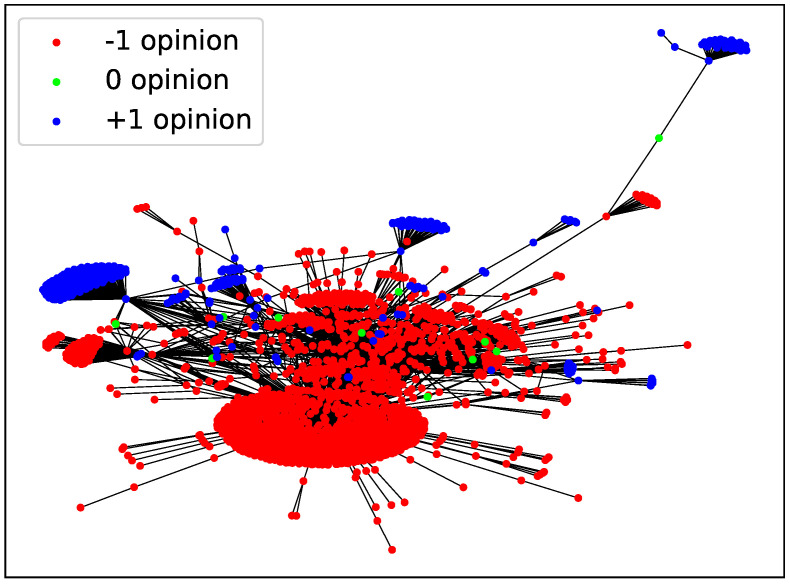
Example of a stationary configuration for #yotambien network at T=0.05 and α=0, reached after $10^{4}$ MMC steps.

**Figure 8 entropy-24-01627-f008:**
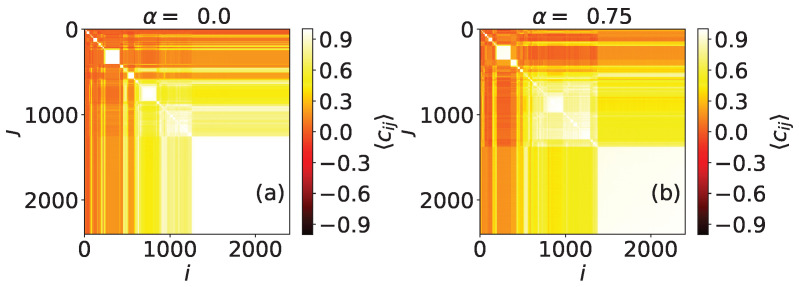
Correlation values for every pair of nodes of the Twitter network #yotambien, obtained using (**a**) α=0 and (**b**) α=0.75. Results are obtained using $10^{4}$ MMC steps, and are averaged over 1000 repetitions.

**Table 1 entropy-24-01627-t001:** Network global coefficients for the explored topologies and average value of the magnetization at low *T* and α=0.5, denoted by $\langle |m{|}_{\circ }\rangle$. *C* is for the clustering, *L* for the average shortest path length, A for the assortativity, and M for the best partition modularity.

	ER z4	#yotambien	#martarovira	#nochebuena	#BA z2
N	2000	2408	29110	20022	BA z2
z	4	2.8	3.67	2.45	2
C	0	0.12	0.08	0.01	0
L	5.58	3.1	4.04	4.14	8.04
A	0	−0.28	−0.27	−0.33	−0.13
M	0.52	0.62	0.65	0.76	0.94
$\langle |m{|}_{\circ }\rangle$	1.0	0.70	0.47	0.32	0.19

## Data Availability

All software and data used to produce the results shown in this paper are publicly available at https://github.com/IreneFerri/Three-state-opinion-model-on-complex-topologies (accessed date: 25 September 2022).

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
