# Peer review of "Three-State Opinion Model on Complex Topologies"

_entropy, 2022, doi:10.3390/e24111627_

Round 1
Reviewer 1 Report
This is a nice paper that studies how a dynamics of a 3-state opinion model is related to the structure of an underlying complex network. These relations have been studied on both synthetic and real-world networks, explaining in which regimes one can observe the formation of consensus, bipartidism or opinion fragmentation.
While the topic is very interesting, the manuscript in its current form lacks precision about the model and more detailed explanations on obtained results. Although the model itself has been introduced in a different publication, some basic information about the model are needed to understand the experiments that were performed.
Several major points to address:
1) In Section 2: it is not defined on which state space the agents live, how are their interactions defined, what are the properties of the underlying networks. Equation (1) has not been fully explained, e.g. how is the sum calculated, how is the multiplication performed.
2) In Section 3.1: authors talk about a critical point, but not about what is this point or how is it obtained.
3) Section 3.2: Why are these particular choices of \alpha made? What happens for \alpha=0.75, as this seems to be an important parameter value afterwards in the paper. Two lines of Fig.1(c) have not been compared. Do we expect a different behavior in Fig.1(b,c) for \alpha not equal 0?
4) How are other parameters chosen? Sometimes T=0.001 is referred to as low temperature, sometimes T=0.1 is called low temperature (Fig.2.). What influence have smaller temperatures in experiment in Fig.2 (since Fig1(b) tells us that for k_{in}=15, already T=0.1 has a big effect)? What happens for larger values of T, e.g. T=0.5?
5) Section 3.3: The network properties are not explained/referred to (how is assortativity defined?), and since they are the basic network properties a question arises: "why are only/exactly these chosen?"
6) Figure 5 #martarovira: how can the shape of curves for alpha=0 vs. alpha =0.5. be explained? In particular the curve for alpha =0 has an unusual form. Where does this come from? How does a curve look like for the alpha = 0.75?
7) How does the type of the underlying network influence the results? Can this methodology be extended to directed graphs (as in the application), or time-evolving graphs?
Author Response
1) In Section 2: it is not defined on which state space the agents live, how are their interactions defined, what are the properties of the underlying networks.”
We replace the first sentence of the section:
“The opinion space in our model is represented by two-dimensional vectors that represent three possible individual opinion states: “
by:
“We model a population of N agents embedded on a generic, undirected and unweighted graph. Each agent can be in one of the following individual opinion states, represented by two-dimensional vectors:”
“Equation (1) has not been fully explained, e.g. how is the sum calculated, how is the multiplication performed.”
After Equation (1) we add the following sentences:
“… extended over nearest neighbors. Therefore every pair of agents connected by an edge contribute to the internal energy of the system. We calculate each contribution as the scalar product of the two opinion vectors and we sum all the contributions to compute the global internal energy (see Fig. 1).”
Additionally, we include a new Figure to illustrate the terms of Equation 1.
We have also replaced the sentence:
“… here temperature being understood as a non-homophilic mechanism for opinion adoption, like social agitation or free-will.”
by:
“… here temperature being understood as a mechanism for opinion adoption different from those captured by Equation 1. While the Hamiltonian rewards that linked agents share the same opinion, the thermal bath gives the agents a certain probability for adopting an opinion that increases the system energy. This behavior can be interpreted as social agitation or free-will.”
2) “In Section 3.1: authors talk about a critical point, but not about what is this point or how is it obtained.”
The tricritical point we mention is calculated in the mean-field approximation in our previous work. We think that it is important to remark its existence, since it is the point that separates the line of continuous transition from the line of discontinuous transitions in the mean-field case. However, we consider that providing further details about its calculation is not relevant for the present work. The reason is that the mean-field approximation is only exact for the fully connected graph, and since most of the topologies studied in the present manuscript are far from a fully connected graph, the outcomes of the dynamics are very different.
3) “Section 3.2: Why are these particular choices of \alpha made? What happens for \alpha=0.75, as this seems to be an important parameter value afterwards in the paper. Two lines of Fig.1(c) have not been compared. Do we expect a different behavior in Fig.1(b,c) for \alpha not equal 0?”
In order to have a more comprehensive vision of the range of parameters, we add the value = 0.75 to the Fig.1(a) (Currently Fig.2(a)).
In the text, we add the following lines to discuss the results obtained for alpha = 0.75:
“For low values of $\alpha$, these networks are not able to achieve global consensus when the number of intra-community links exceeds $8$ (however the modules reach internal consensus in all cases, see. Figure~\ref{z15}(a)).
When $\alpha = 0.75$, we observe that $\langle |m| \rangle \neq 1$ even for $k_{in} \le 8$.
We speculate that these results are related to a putative first order transition analogous to that of the fully-connected graph \cite{Ferri_2022}, which may occur at alpha around the interval $\alpha \in (0.8, 1)$. In this range the dynamics end up in consensus, but this consensus occurs in a polarized opinion for roughly half of the simulations, and in a neutral opinion the other half. For $k_{in} > 8$ we observe a decay in $\langle |m| \rangle$, similar to the one presented for $\alpha = 0$. Finally, for $\alpha \ge 1$, when the neutral consensus becomes an absolute energy minimum, we observe clear preference for the neutral consensus for all $k_{in} $. Nevertheless, some polarized clusters can appear for $\alpha = 1$, especially for the largest values of intra-community links.”
Regarding (current) Fig. 2(b) and Fig. 2(c), we expect that for alpha close to the first order transition (but below 1) all networks have more prevalence of neutral clusters or neutral consensus at low temperatures. However, the figures aim to characterize the behavior of <|m|>, the prevalence of polarized opinions in the limit case of neutrality parameter alpha = 0. In the low alpha limit is where the temperature plays a more important role in the value of <|m|>.
4) “How are other parameters chosen? Sometimes T=0.001 is referred to as low temperature, sometimes T=0.1 is called low temperature (Fig.2.). What influence have smaller temperatures in experiment in Fig.2 (since Fig1(b) tells us that for k_{in}=15, already T=0.1 has a big effect)? What happens for larger values of T, e.g. T=0.5?”
For the network considered in (current) Fig.3 we observe no difference in the outcomes between both temperatures T* = 0.001 or T* = 0.1, just the simulations stabilize faster when T* = 0.1. in (current) Fig. 2(b) there is noise in the data for low temperatures. This noise comes from the fact that for kin = 15 the are 3 possible outcomes when alpha is low (as shown in current Fig. 3(b)). Two of them correspond to consensus in both polarized opinions and the third is for three communities in local consensus in one extremist opinion and the fourth in the opposite opinion state (<|m|> = 0.5 and n0 = 0). Furthermore, the latter outcome has degeneracy 4. The results were averaged over 100 repetitions, so the statistics were poor in this range of parameters. For this reason we have performed more repetitions for the Newman-Girvan networks with k_{in} = 15.
For T* = 0.5 and alpha below the first order transition line, the behavior is very similar to the one observed for alpha = 0 (see current Fig2(b)). For bigger alpha we expect a prevalence of neutral opinion, as we mention in point 3).
5) “Section 3.3: The network properties are not explained/referred to (how is assortativity defined?), and since they are the basic network properties a question arises: "why are only/exactly these chosen?”
We explore the values for this particular network properties because they are features that a priori could be related to a fragmented situation, with coexistence of opinion clusters instead of consensus at low temperatures.
For instance, assortativity is a global feature of the network that is calculated as the Pearson correlation coefficient of degree between pairs of linked nodes. Positive values of this coefficient indicate a correlation between the degrees of linked nodes, while negative values indicate that low degree nodes are more likely to be connected to high degree nodes. It is commonly used to classify networks (technological networks have a low Pearson correlation coefficient while social networks are very assortative). We presumed that it could be relevant to explain the behavior of the order parameters, but finally it is not the case.
6) “Figure 5 #martarovira: how can the shape of curves for alpha=0 vs. alpha =0.5. be explained? In particular the curve for alpha =0 has an unusual form.”
We have added the following two paragraphs in this section in order to better explain the results for real networks, and #martarovira in particular:
"The highest value for the magnetization at low temperatures is found at $\alpha$ around $0.5$; $\alpha = 0.75$ in the case of BA graphs with $\langle k \rangle = 2$. When the magnetization reaches its maximum value the fraction of neutral agents $n_0 $ is zero. From a social point of view, this means that the system gets divided into communities formed by positive or negative agents, but there is a stronger majority in one polarized opinion than for other values of $\alpha$. Neutral communities are found at higher values of $\alpha$, destabilizing the polarized majority. The fact that a moderate neutral intensity catalyzes majority in one polarized opinion adds to the conclusions presented in \cite{Svenkeson2015}.
The networks \#martarovira and \#nochebuena exhibit a non-monotonic behavior of $\langle |m| \rangle$ with $T$ for $\alpha = \{0, 0.5 \}$. This is caused by the fact that the model presents a very complex energy landscape with multiple local minima when embedded in this topologies. This landscape is sensitive to the neutrality parameter $\alpha$, since it changes the energy difference between the local neutral consensus and local polarized consensus. When the temperature increases, some energetic barriers can be easily overcome, so the attractors of the dynamics change. In some cases this can enhance the dominance of one polarized opinion over the other one, which corresponds to higher average absolute magnetization, in others it is vice versa."
“Where does this come from? How does a curve look like for the alpha = 0.75?”
This curve is very close to the alpha = 0.5 values, so we remove it from the figure for clarity.
7) “How does the type of the underlying network influence the results?”
We find that modularity is the network feature that influences the most the dynamics results. When the network has very well-defined communities global consensus becomes impossible and, for low alpha, only polarized bipartidist scenarios appear at low temperatures.
Can this methodology be extended to directed graphs (as in the application), or time-evolving graphs?
We are thankful for this suggestion. Indeed, the model could be extended to directed graphs, but it is beyond the scope of this work. Regarding to temporal graphs we are already exploring these graphs by applying the model to agents moving with velocity v on a plane. Here, agents interact only if they are physically closer than a given distance.
——————————————————————————————
—————————————————————————————
Other small changes:
- In line 110 the formula was misprinted and has been replaced by .
- In (current) Fig. 2 label changes according to the previous point.
- Other minor changes throughout the manuscript.

Reviewer 2 Report
As I understand, this paper tests the already elaborated model on those network topologies, on which the model was not tested before.
I believe that the paper makes a contribution to the field of opinion dynamics. Perhaps, the results are not too impressive, but, nonetheless, the paper could be a nice contribution to the Special Issue.
However, at the moment, the manuscript has one sound weakness: poor presentation of the model. I mean, Section 2 needs substantial improvement. More precisely, I find it difficult to understand how agents communicate, how they change opinions, and how the neutrality parameters \alpha affects opinion dynamics processes. Perhaps, the authors could argue that the description of the model can be found in Ref. [31] (which is not, actually, published - on this basis I recommend replace this reference with the link to arxiv paper, if it exists; otherwise, this Ref. should be removed, in my opinion). Nonetheless, my point of view is that potential readers of the manuscript should have an opportunity to look at detailed presentation of the model without switching to other papers.
Further, I suppose that the authors should carefully define such terms as "thermal bath", "non-homophilic mechanism of opinion adoption".
Good luck!
Author Response
“As I understand, this paper tests the already elaborated model on those network topologies, on which the model was not tested before.
I believe that the paper makes a contribution to the field of opinion dynamics. Perhaps, the results are not too impressive, but, nonetheless, the paper could be a nice contribution to the Special Issue.
However, at the moment, the manuscript has one sound weakness: poor presentation of the model. I mean, Section 2 needs substantial improvement. More precisely, I find it difficult to understand how agents communicate, how they change opinions, and how the neutrality parameters \alpha affects opinion dynamics processes.”
After Equation (1) we add the following sentences:
“… extended over nearest neighbors. Therefore every pair of agents connected by an edge contribute to the internal energy of the system. We calculate each contribution as the scalar product of the two opinion vectors and we sum all the contributions to compute the global internal energy.”
Additionally, we include a new Figure to illustrate the terms of the Equation 1.
Perhaps, the authors could argue that the description of the model can be found in Ref. [31] (which is not, actually, published - on this basis I recommend replace this reference with the link to arxiv paper, if it exists; otherwise, this Ref. should be removed, in my opinion).
We have uploaded Ref [31] to arxiv (https://arxiv.org/abs/2210.03054). We add the link in the references.
Nonetheless, my point of view is that potential readers of the manuscript should have an opportunity to look at detailed presentation of the model without switching to other papers.
Further, I suppose that the authors should carefully define such terms as "thermal bath", "non-homophilic mechanism of opinion adoption”.
Good luck!
In section 2, we have replaced the sentence:
“… here temperature being understood as a non-homophilic mechanism for opinion adoption, like social agitation or free-will.”
by:
“… here temperature being understood as a mechanism for opinion adoption different from those captured by Equation 1. While the Hamiltonian rewards that linked agents share the same opinion, the thermal bath gives the agents a certain probability for adopting an opinion that increases the system energy. This behavior can be interpreted as social agitation or free-will.”
———————————————————————————————
——————————————————————————————--
Other small changes:
- In line 110 the formula was misprinted and has been replaced by .
- In (current) Fig. 2 label changes according to the previous point.
- Other minor changes throughout the manuscript.

Round 2
Reviewer 1 Report
Authors have addressed my comments.
Reviewer 2 Report
The authors have performed a considerable work to meet my and other reviewer's comments.
Still, I suppose that the presentation of the manuscript can be improved, but its current level tends to be acceptable.
One minor comment is about Ref. [42]: it seems like this reference should be corrected.
The paper looks publishable, and would not require another round of reviews (unless the Editor deems it necessary).